# Exosomes: Versatile Nano Mediators of Immune Regulation

**DOI:** 10.3390/cancers11101557

**Published:** 2019-10-14

**Authors:** Qi Li, Helei Wang, Hourong Peng, Ting Huyan, Nicholas A. Cacalano

**Affiliations:** 1Key Laboratory for Space Bioscience and Space Biotechnology, School of Life Sciences, Northwestern Polytechnical University, 127 YouyiXilu, Xi’an 710072, Shaanxi, China; liqi_1111@nwpu.edu.cn (Q.L.); penghourong@mail.nwpu.edu.cn (H.P.); huyanting@nwpu.edu.cn (T.H.); 2Department of Gastrointestinal Surgery, the First Hospital of Jilin University, Changchun 130021, Jilin, China; helei@jlu.edu.cn; 3Department of Radiation Oncology, David Geffen School of Medicine at UCLA, Los Angeles, CA 90095, USA

**Keywords:** exosome, immunocyte, immunoregulation, immunotherapy, natural killer cells

## Abstract

One of many types of extracellular vesicles (EVs), exosomes are nanovesicle structures that are released by almost all living cells that can perform a wide range of critical biological functions. Exosomes play important roles in both normal and pathological conditions by regulating cell-cell communication in cancer, angiogenesis, cellular differentiation, osteogenesis, and inflammation. Exosomes are stable in vivo and they can regulate biological processes by transferring lipids, proteins, nucleic acids, and even entire signaling pathways through the circulation to cells at distal sites. Recent advances in the identification, production, and purification of exosomes have created opportunities to exploit these structures as novel drug delivery systems, modulators of cell signaling, mediators of antigen presentation, as well as biological targeting agents and diagnostic tools in cancer therapy. This review will examine the functions of immunocyte-derived exosomes and their roles in the immune response under physiological and pathological conditions. The use of immunocyte exosomes in immunotherapy and vaccine development is discussed.

## 1. Introduction

Extracellular vesicles (EVs) are a heterogeneous family of lipid bilayer-derived nanovesicles that are released by almost all living cells [1,2]. Exosomes, which represent one subpopulation of EVs, arise from a unique biogenesis pathway and are characterized by a cup-shaped morphology under electron microscopy a diameter of 30–100 nm and a density of 1.13–1.19g/mL [3].

The biogenesis of exosomes is a highly dynamic but ordered process. By inward budding of plasma membrane, membrane-enclosed compartments called early endosomes (EEs) are created [4,5,6]. Subsequently, inward budding of EE membranes generates intraluminal vesicles (ILV) [7], a process that is mediated by the endosomal sorting complex required for transport (ESCRT), tetraspanins, and the lipid lysobisphosphatidic acid (LBPA) [8]. ILVs are loaded with cargo by capture during vesicle formation or via a trans-golgi process regulated by CD2AP and LMAN2 [9] and mature into late endosomes, or multivesicular bodies (MVB) [10], which fuse with either lysosomes for content degradation or the plasma membrane to release exosomes into the extracellular environment [11,12,13]. Thus, the composition of exosomes is generally recognized as representative of their parental cells and they are utilized as biomarkers of cellular function in vivo [14]. Although exosomes are characterized primarily as vehicles for the elimination of cellular waste, they are also active players in diverse cellular functions [15], especially in the context of the immune system. Critically, exosomes are often enriched in molecules associated with specific biological functions that can affect cells at distal sites after release into the circulation. Exosomes transmit information and activate biological responses in target cells through several potential mechanisms: 1. Direct fusion with the plasma membrane and release of exosomal contents, 2. The uptake of intact exosomes into endosomes and subsequent release of contents into the cytoplasm, and 3. Juxacrine signaling between ligands expressed on exosomes and cognate receptors on target cells, without intracellular delivery of cargo [16,17].

The process of target cell recognition, uptake, and release of cargo is under investigation and the molecular components are currently being identified and validated. The recognition and binding of target cells by exosomes is not completely understood and it may be a largely non-specific stochastic process, although there is evidence for the preferential uptake of certain exosomes by specific cell types. Following initial contact, exosomes establish a point of entry into acceptor cells. This stage might depend on relatively non-specific mechanisms such as macropinocytosis or micropinocytosis or may be dependent on specific receptor-ligand interactions [13,17]. Several families of proteins have been identified on the surface of exosomes,, such as T and B cell receptors, cytokines and cytokine receptors, integrins, and lectins, which may provide specificity to this process. Following exosome internalization, which is mediated through several different pathways, the contents of exosomes are released by either direct fusion with the plasma membrane or via uptake as intact vesicles via the endosomal pathway [17]. This distinction is particularly important in the context of antigen presenting cells (APCs), which can process exosomal antigens through endosomes where they are loaded onto MHC molecules and presented to effector lymphocytes, activating the immune response.

While the molecular nature of exosome-target cell interactions is still incompletely understood, these findings suggest that at least some components of the recognition and entry process are receptor-dependent and thus can be manipulated to increase the precision of exosome targeting. This is of great interest therapeutically, as exosomes can be engineered genetically and pharmacologically to express receptors that target specific cell types or tissues, which would enhance the specificity of their target cell binding, increase their efficacy, and reduce deleterious off-target effects [18].

Through the processes of biogenesis and capture by acceptor cells, exosomes deliver proteins, lipids, and a number of RNA species, including regulatory micro RNAs (miRNAs) from their cells of origin, all of which are capable of modulating target cell function and gene expression [19,20,21,22,23].

Many studies have shown that immunocytes secrete exosomes [3,24], which act as immune modulators in normal and disease conditions. Here, we review recent findings and highlight the biological and pathophysiological functions of immunocyte-derived exosomes in immunoregulation as well as their use in immunotherapy and vaccine development. We use the term exosome to describe all vesicle structures of standard size (30–100 nm) that express typical exosomal markers.

## 2. Exosomes: Efficient Nano-Messengers of Antigen Presentation

In 1996, exosomes were first identified as nanovesicles that are enriched in major histocompatibility complex class II (MHC-II) molecules secreted from B-lymphoblastoid cells [25]. Subsequently, exosome-like structures were found to be associated with all types of immunocytes, including B and T lymphocytes, macrophages, dendritic cells (DCs), natural killer (NK) cells, mast cells, and thymocytes [3,24,26]. Molecular profiling of exosomes revealed the classical exosome markers, such as intraluminal heat shock proteins, tetraspanins, multivesicular body (MVB) biogenesis proteins, as well as lipid-related proteins and phospholipases, which are associated with exosome biogenesis [27,28,29]. Exosomes that are derived from immunocytes, however, were enriched in proteins with immunological function, including antigen presenting molecules (MHC class I, MHC class II, and CD1), adhesion molecules (CD11b, CD54/ICAM-1), and costimulatory proteins (CD86) [27,30]. The molecular profiles of exosomes are thus similar to but distinct from those of their parental cells, which suggests that they are designed to perform specialized biological functions [8]. Indeed, exosomes that are derived from APCs and other immunocytes can present antigens to T cells and activate immune responses via classical MHC-restricted mechanisms, involving multiple cell-cell interactions. Exosomes function in initial T cell priming, differentiation of mature T cells, and the development of effector functions. Antigen presentation by exosomes can be accomplished by transfer of MHC-peptide complexes between naïve dendritic cells, uptake and presentation of exosome cargo by mature DCs, as well as by direct T cell activation without the need for uptake and further processing by APCs.

In addition to unique sets of proteins, the lipid composition of exosomes is also distinct from other cellular membranes. Exosomal membranes are enriched in certain lipids, such as sphingomyelin, lysophosphatidylcholine, and saturated fatty acids, but relatively poor in others, such as phosphatidylcholine and diacylglycerol [31]. Cholesterol and ganglioside GM3 in exosome membranes confer excellent detergent-resistance and the capacity for extended survival in the extracellular environment [32]. In addition, proteins that protect them from degradation enhance the stability of exosome membranes. Two molecules that inhibit complement-dependent membrane attack and cell lysis, CD55 and CD59, were both found at high levels on exosomes that were derived from human APCs and thus are likely to protect exosomes from complement-mediated destruction in the plasma [33]. Likewise, the anti-phagocytic protein CD47 has been found on the surface of exosomes from multiple cell types, including T lymphocytes. CD47 blocks macrophage-mediated uptake of exosomes by engaging the Sirp-α inhibitory receptor, sending a “do not eat” signal, and preventing destruction by phagocytosis [18]. Liposomes and exosomes lacking CD47 displayed reduced retention in the circulation relative to CD47+ exosomes, which suggests that the enhanced in vivo stability of exosomes is due in part to CD47-dependent inhibition of phagocytosis. The combination of specialized, biologically active components and physical stability suggests that immunocyte exosomes are uniquely suited as vectors for the transport of cargo, such as tumor antigens or therapeutic drugs through the circulation to target cells in vivo.

B cells capture cell-free antigens via cell surface B-cell receptors (BCRs) and translocate them into the endosomal pathway for processing in antigen presentation or release them as exosomal cargo [34,35,36]. Recent studies have shown that B cell exosomes activate the immune response through multiple mechanisms. They express high levels of MHC-I, MHC-II molecules, co-stimulatory molecules and human leukocyte antigens (HLA), and induce antigen-specific T cell responses [25,37,38,39]. B-cell exosomes were also found to contain integrins and ICAM-1 (CD54) that promote anchoring to the extracellular matrix (ECM), and facilitate delivery of cargo to target cells through heterotypic interactions with leukocyte function-associated antigen (LFA)-1 on lymphocytes. Hence, B cell exosomes might penetrate tissues via interaction with the ECM and deliver immune cargo to pro-inflammatory effector cells through specific receptor-ligand interactions [40]. Likewise, CD38, a glycoprotein with glycohydrolase activity, was detected on human lymphoblastoid B cell-derived exosomes. This protein is associated with membrane rafts in lymphocytes that contain signaling and co-stimulatory proteins, such as CD81 and the Src-family kinase lyn, suggesting that they can act as intercellular messengers of lymphocyte activation [41]. In a related study, Papp, et al. [42]. reported that murine macrophages and B cells covalently fix C3 complement fragments to their cell membranes and release them on exosomes. The engagement of exosome-bound C3+ by G protein-coupled receptor C3aR on T cells enhanced T-cell responses in the presence of sub-threshold antigenic stimulation [42].

Dendritic cells (DCs) are the most potent APCs, and they can release up to one million MHC-II molecules in 24 h via the DC-derived exosome (Dex) pathway [43]. Dex contain the machinery for antigen presentation and they have been shown to activate T cell-mediated cytotoxicity through engagement of T cell receptor (TCR) complexes, while adhesion molecules aid in targeted transfer of exosomal cargoes to lymphocytes via interactions with LFA-1 [44]. Because of their stability in the extracellular environment, Dex can activate T cells at distal sites, in the absence of intact APCs [45]. Segura, et al. reported that CD8+ DCs express high levels of LFA-1 and are therefore likely to internalize Dex in vivo [46]. Thus, Dex-mediated antigen transfer might promote both MHC II-dependent classical presentation by DCs as well as MHC I-dependent “cross-presentation” of exogenous antigens by CD8+ DCs. Likewise, CD4+ T cells are also able to engage MHC-II-containing Dex via a LFA-1-dependent mechanism [47], which suggests that Dex potentiate both T cell help and the development of cytotoxic effector T cell function. These studies also demonstrated that Dex could acquire antigens either through internalization and processing of antigens by DC or by direct capture of free antigens [48]. However, recent findings suggest that prior processing of antigens is required for optimal exosome-mediated T cell activation [49].

Exosomes have also been shown to participate in more complex communication among components of the immune system. For example, Dex are a danger signal to the immune system in cases of pathogen infection. Through capture and presentation of bacterial toll-like receptor ligands (TLR-Ls), Dex stimulated bystander DCs to secrete tumor necrosis factor (TNF)-α and proinflammatory cytokines, which in turn induced IFN-γ secretion by NK cells [50], activating the innate immune response. A recent study demonstrated that T cells and APCs could reciprocally exchange exosomes. T cell exosomes transferred miRNAs, such as miR-335 to DCs at the immune synapse, potentially modulating APC gene expression [51]. Furthermore, regulation of the immune response by exosomes likely involves interactions between multiple immune cell subpopulations. A recent study indicated that, in vivo, the activation of cytotoxic T lymphocytes (CTLs) by B cell-derived exosomes was dependent on the presence of CD4 T cells and NK cells, which suggests that exoxome-driven immune responses require T cell help and cross-talk between immunocytes [52]. Likewise, it was shown that immune stimulation by Dex was dependent on cross talk between CD4+ T cells and B cells. Such complex interactions must be considered when using Dex as immune response modifiers in clinical applications [53].

According to recent studies, at least 98 immunocyte derived molecules have been found in human exosomes and 43 immunocyte-derived proteins have been detected on murine immunocyte exosomes with the potential to modulate immunological functions and regulate exosome stability and trafficking in vivo. Table 1 and Table 2, and Figure 1 summarize our current knowledge of immune-related molecules found in exosomes.

## 3. NK Cell-Derived Exosomes, Novel Cytotoxic Immune Mediators

NK cells, a subset of large granular lymphocytes, lie at the crossroads of innate and adaptive immunity. They kill transformed and virally-infected cells without prior immunization, and stimulate adaptive immunity by secreting pro-inflammatory cytokines and chemokines [80]. A recent study showed that, following activation, NK cells released exosomes that contained cytotoxic proteins, such as perforin, granulysin, and granzymes A and B, and mediated tumor cell lysis by activating caspase pathways, in a manner similar to intact NK cells [69]. Another study identified FasL and perforin on NK cell-derived exosomes, which displayed cytotoxic activity against several tumor cell lines [70]. Likewise, perforin, FasL, and TNF-α were detected on exosomes that were derived from the human NK92 cell line, which promoted cytolysis of melanoma cells in vitro and in vivo [71]. Presensitization may help to improve the cytotoxic activity of NK cell exosomes. Prior exposure of NKs to neuroblastoma cells increased the secretion of exosomes expressing NK receptors CD56, NKG2D, and KIR2DL2, and displayed greater cytotoxicity against neuroblastoma tumors relative to exosomes from naïve NK cells [81]. These studies suggest that NK cell exosomes might be used as immunotherapeutics for the treatment of tumors and viral diseases (Figure 2).

## 4. Exosomes—Nano regulators of Immune Response with Two Faces

Immunocyte exosomes can stimulate or attenuate immune responses, depending on cellular context, molecular composition, and the microenvironment in which they are produced (Figure 3). For example, exosomes from mature (mDex) and immature DCs (imDex) have opposing effects on the immune system. In several studies, immature DCs expressed low levels of co-stimulatory molecules and promoted immune tolerance, whereas mature DCs expressed high levels of MHC class II, B7.2, ICAM-1 (CD54), and CD80, reduced levels of MFG-E8 relative to imDex, and had greater potential to drive antigen-specific T cell proliferation and in vivo immune responses [57,72,82].

Following the treatment of immature DCs with immunomodulatory cytokines IL-10 and IL-4, the resulting imDex impaired delayed-type hypersensitivity (DTH) and alleviated the severity of collagen-induced arthritis (CIA) in a murine footpad model [83]. In a rat model of intestinal transplantation, imDex suppressed the alloreactive T-cell response via up-regulation of IL-10 levels, and prolonged the survival of intestinal allografts [84]. In combination with the immunosuppressant rapamycin, imDex inhibited T-cell activation and prolonged the survival of cardiac allografts in vivo, which was associated with increased splenic CD4+CD25+ regulatory T (Treg) cells and diminished anti-donor antigen responses [85].

Likewise, Treg-derived exosomes have been shown to maintain immune tolerance. In a rat model of kidney transplantation, Treg-derived exosomes suppressed T cell proliferation, delayed allograft rejection, and prolonged allograft survival [86]. In a related study, exosomes that were derived from CD4+25+ and CD8+25+ Tregs repressed CD8+ T cell mediated immunity against B16 melanoma cells both in vitro and in vivo, indicating a negative function for Treg exosomes in antitumor immune responses [79]. These studies suggest that immunocyte-derived exosomes might have deleterious effects on the immune response to infectious agents and cancer. However, from a therapeutic standpoint, their immunosuppressive properties may be exploited for the treatment of chronic inflammation, autoimmune disease, or allograft rejection.

## 5. Remodeling the Composition and Biological Function of Immunocyte Exosomes: Implications for Immunotherapy

Extracellular ligand stimulation has been shown to alter the secretion, morphology, and biological function of immunocyte exosomes. Saunderson, et al. found that CD40 and IL-4 stimulation of human and murine B cells induced the production of exosomes that were enriched in MHC-I, MHC-II, and CD45RA (B220), as well as components of the BCR complex [54]. Likewise, TCR crosslinking induced the production of T cell exosomes that expressed high levels of signaling components, such as tyrosine kinases of the Src family, c-Cbl, and TCR/CD3 complexes [64].

Blanchard, et al. showed that exosomes from IL-12-stimulated CTLs were larger than normal and enriched in immunoregulatory molecules, such as Zap70, Granzyme B, STAT3, and STAT5, which directly activated bystander naive CD8+ T cells to increase interferon-γ (IFN-γ) and granzyme B (GZB) levels in the absence of antigen [87]. In a similar study, exosomes from IL-2-stimulated CD3+ T-cells promoted the proliferation of autologous resting CD3+ T cells in the absence of accessory cells [88]. DCs also responded to activation by stimuli such as LPS by increasing the secretion of Dex [65]. Interestingly, after LPS stimulation, exosomes secreted by mature DCs were 50- to 100-fold more likely to induce antigen-specific T-cell activation than imDex in vitro [73].

In addition to an activating role in adaptive immunity, APCs can promote immune tolerance through the secretion of exosomes with altered levels of antigen presenting and costimulatory molecules. Ruffner, et al. reported that IL-10 treatment significantly down-regulated the surface expression of MHC-I, MHC-II, and costimulatory molecules B7-1, B7-2, PD-L1, and PD-L2 on DCs. Exosomes derived from IL-10-treated DCs suppressed DTH responses by interfering with T cell costimulation [74].

The studies that are cited above demonstrate that ex vivo treatment with cytokines or other stimuli can shape the molecular and functional repertoire of immunocyte exosomes, tailoring them for specific functions (Figure 3). Further study is needed to understand how to best manipulate the production and composition of exosomes to maximize their clinical potential.

## 6. Modified Exosomes, a More Efficient Vaccine or Immunosuppressive Strategy in Immunotherapy

Strategies for improving the efficacy of exosome-based immunotherapeutics include (1) direct loading of antigens onto exosomes ex vivo or (2) genetic or pharmacologic modification of APCs to engineer exosome composition, improve their yield, and increase their biological activity [52,83,89]. Aline, et al. demonstrated that Dex loaded with toxoplasma gondii antigens accumulated in the spleens of mice, elicited a Th1-dependent toxoplasma-specific immune response, and conferred protection against a secondary challenge with toxoplasma [90]. Similarly, Colino, et al. showed that exosomes from DCs that were loaded with diphtheria toxoid (DT) induced a type 1 (IgG2a and IgG2b) DT-specific response in vivo [91]. In two recent studies, IL-4 or major birch allergen Betv1—a T-cell-activating peptide—were expressed in APCs, and the resulting exosomes inhibited the activity of APC and T cells [92] or induced T-cell proliferation and synthesis of T(H)2-like cytokines (IL-5 and IL-13) in vitro [55], respectively.

Similarly, tumor-specific antigenic peptides as well as tumor cell lysates have been used to modify exosomes for the purpose of eliciting antitumor immunity [30,58,93]. In one study, MHC class I and class II-restricted MART1 tumor peptides loaded onto Dex in acidic conditions bound to MHC molecules with high affinity and activated CD8+ CTL and CD4+ T-cell responses after processing by DCs [94]. Similarly, Dex derived from DCs loaded with lysates from chaperone-rich cells (CRCLs) and GL261 murine glioblastoma tumor cells induced proliferation and CTL activity of CD4+ and CD8+ T cells and enhanced the production of IL-2 and IFN-γ in vitro. In vivo, the Dex inhibited tumor growth and prolonged the survival of tumor-bearing mice [95]. In a murine model of hepatocellular carcinoma (HCC), exosomes that were derived from α-fetoprotein (AFP)-expressing DCs (DEXAFP) decreased the levels of CD25+ Foxp3+ Tregs and anti-inflammatory cytokines IL-10 and TGF-β, while increasing CD8+ CTLs and proinflamatory cytokines IFN-γ and IL-2 in the tumor microenvironment, resulting in prolonged survival of tumor-bearing mice [96]. Rountree, et al. [97] developed an exosome-based anti-prostate cancer vaccine—MVA-BN-PRO—that contained two chimeric proteins derived from tumor-associated antigens, prostate-specific antigen (PSA), and prostatic acid phosphatase (PAP) fused to the C1C2 domain of lactadherin, targeting them specifically to the exosome. The vaccine showed increased immunogenicity and therapeutic efficacy in a PAP-expressing murine prostate cancer model [97]. Gehrmann, et al. [98] loaded bone marrow-derived dendritic cell (BMDC) exosomes with α-galactosylceramide (αGC), an invariant NKT (iNKT) receptor ligand, in combination with the model antigen ovalbumin (OVA), to activate iNKT immune cells. Exosomes that were pulsed with both αGC and soluble OVA synergistically induced NK and γδ T-cell innate immune responses, and also amplified OVA specific T- and B-cell responses. In the B16/OVA murine model of melanoma, αGC/OVA exosomes decreased tumor growth, increased antigen-specific CD8+ T-cell tumor infiltration, and increased median survival [98]. Likewise, Dex that were generated from DCs overexpressing ovalbumin (OVA-Dex) stimulated naive CD8+ T cell proliferation and differentiation into CTL and induced antigen-specific IgG production [99]. In a related study, OVA-Dex were used to “immunize” DC via exosome uptake. Dendritic cells immunized with OVA-Dex induced OVA-specific CTL responses, antitumour immunity, and CD8+ T-cell memory more efficiently than control exosomes from DCs expressing OVA and more effectively eradicated OVA^+^ transplanted tumors [75]. Similarly, CD4+ T cells were immunized with OVA-Dex, and exosomes derived from the immunized CD4+ T cells stimulated a long-term, OVA-specific CD8+ T cell memory response, and immunity against melanoma cells in vitro and in vivo [100].

Recently, Anticoli, et al. [101]. developed a novel, rapid method of loading antigens into exosomes by injecting a DNA vector encoding the antigen of interest fused to the C-terminus of an exosome-anchoring protein into autologous APCs. Antigens of different origins and sizes were efficiently uploaded into exosomes while using this approach [101]. These studies illustrate the progress made toward the use of exosomes in immunotherapy through technical advances that increase their efficacy in stimulating immune responses.

Conversely, other studies have focused on modifying exosomes as an immunosuppressive strategy for treating autoimmune diseases or prolonging the survival of allografts. Dex derived from DCs infected with a Fas-L recombinant adenovirus suppressed antigen-specific immune responses and DTH inflammatory responses in a collagen-induced murine footpad arthritis model [102]. Similarly, exosomes from Fas-L+ EBV-transformed B cells induced tolerance by Fas-mediated apoptosis of antigen-specific T helper (Th) lymphocytes [103].

Exosomes from BMDCs that were treated with the anti-inflammatory drug atorvastatin up-regulated the levels of indoleamine 2,3-dioxygenase (IDO)/Tregs and suppressed autoimmune responses in rat model of experimental autoimmune myasthenia gravis (EAMG)[104]. The immunosuppressive effect of exosomes was partially reversed by pre-treatment with an anti-FasL blocking antibody, indicating the involvement of the Fas pathway. Similarly, TGF-β1-loaded Dex impaired Ag-specific Th1 and IL-17 responses and promoted IL-10 production and the generation of Tregs in murine models of experimental allergic encephalomyelitis (EAE) and inflammatory bowel disease (IBD), respectively [105,106].

Dex-immunized T cells also have immunosuppressive effects. Zhang, et al. demonstrated that exosomes from OVA-Dex-immunized CD4+ T cells directly inhibited CD4+ and CD8+ T responses [78]. Likewise, exosomes derived from CD8+ T cells expressed both OVA-specific T-cell receptors (TCRs) and Fas ligand. The CD8+ T cell exosomes inhibited CD8+ CTL responses as well as antitumor immunity against OVA-expressing B16 melanoma cells [107]. These studies suggest that the outcome of exosome-based vaccines is likely to be context, antigen, and cell-type-dependent. Exosomes can be engineered to stimulate or suppress the immune response by careful pharmacologic or genetic manipulation.

## 7. Clinical Application of Exosome-Based Vaccines

In a clinical trial of stage III/IV melanoma, Escudier, et al. [108]. produced a Dex-based vaccine by loading tumor antigen MAGE3 peptides onto exosomes that were derived from autologous DCs. The Dex vaccine was administered to fifteen patients at two doses with no detectable toxicity, demonstrating the feasibility of large-scale exosome production and the safety of exosome therapy [108]. In a phase I clinical study, autologous MAGE-loaded Dex were administered to 13 patients with advanced non-small cell lung cancer (NSCLC). Dex therapy was well tolerated and a subset of patients experienced long-term stability of disease and activation of immune effectors, including NK cells and MAGE-specific T cells [109]. Further, in a Phase III study where an autologous Dex vaccine (IFN-γ-Dex) was given to 22 NSCLC patients, median survival was increased and 32% of patients experienced disease stabilization of more than four months. A subset of treated patients displayed enhanced NKp30-dependent NK function. Increased NK function and longer progression-free survival in patients correlated with the expression levels of NKp30 and MHC-II on IFN-γ-Dex vaccine [110]. In another Phase I clinical trial, autologous, immunocyte-derived exosomes from ascites were combined with granulocyte-macrophage colony-stimulating factor (GM-CSF) and given to 40 patients with advanced colorectal cancer. This combination therapy induced beneficial tumor-specific antitumor cytotoxic T lymphocyte (CTL) responses [111].

When compared to vaccines using intact DCs, clinical trials using exosomes have been limited in scope, and their potential for eliciting anti-tumor immunity has not been fully evaluated. Whereas most attempts to use immunocyte exosomes clinically have focused on ex vivo purification of vesicles from autologous cells, recent studies indicate that exosome-induced immune responses may in fact be MHC-independent [112]. In addition, as cancer patients often display reduced numbers of circulating immune cells as well as impaired lymphocyte and DC function, autologous immunocytes might be suboptimal sources for the production of exosome-based therapeutics. In light of these findings, Samuel and Gabrielsson have proposed that allogeneic exosomes may be more effective, as they are able to induce robust and wide-ranging anti-tumour immune responses in vivo [113].

Accordingly, studies are underway directly comparing the relative efficacy of intact DCs and Dex in stimulating anti-tumor immune responses. Currently, there are at least six ongoing or recently completed clinical trials using immunocyte exosomes as vaccines or biomarkers of immune function in cancer.

## 8. Roles for Immunocyte Exosomes in Pathophysiologic Processes, Inflammation, Tissue Remodeling, and Response to Injury

In addition to their antigen presentation functions, immunocyte exosomes have been shown to regulate tumor growth and metastasis, leukocyte trafficking and inflammation, and the pathogenesis of infectious diseases (Figure 3). T cell-derived exosomes have been shown to promote the invasion and metastasis of melanoma and lung cancer by increasing matrix metalloproteinase-9 (MMP-9) expression through Fas/FasL-dependent ERK and NF-kB activation [114]. Interestingly, the secretion of Fas-L+ T cell-derived exosomes was controlled by diacylglycerol kinase-α (DGK-α), suggesting that pharmacologic regulators of DGK-α can modulate the biosynthesis and release of toxic exosomes in vivo [66,115]. Likewise, Yang, et al. found that oncogenic miRNAs in exosomes derived from tumor-associated macrophages were delivered to breast cancer cells and enhanced their invasive properties in vitro [68].

In the tumor microenvironment, exhausted CD8+ T cells (PD1+TIM3+) secreted exosomes, which impaired normal CD8+ T cell proliferation, activation, and the production of cytokines, and may interfere with antitumor immunity [116]. On the other hand, CTL-derived exosomes have also been shown to enhance the activation of low-affinity CTLs that play an important role in controling of intracellular pathogens and cancers [117]. Lymphocyte exosomes can either activate or impair antitumor immunity, depending upon cellular context and the conditions of the tumor microenvironment.

Macrophage exosomes have also been shown to have tumor-promoting effects. Lan, et al. demonstrated that M2-polarized macrophages could promote colon cancer invasion and metastasis through the delivery of micro RNAs miR-21-5p and miR-155-5p that target the antimetastatic gene BRG1 [118]. In contrast, others have shown that exosomes from M1 polarized macrophages induced exit of breast cancer stem cell from quiescence and increased their migratory capacity, whereas M2 macrophage exosomes maintained cancer stem cell dormancy [119]. These data suggest that exosomes from M1 and M2-polarized macrophages differentially regulate cancer stem cell mobilization and epithelial to mesenchymal transition (EMT). Biological targeting agents that alter the M1/M2 balance may therefore reprogram macrophage exosomal function in the tumor microenvironment to favor inhibition of EMT and metastasis.

Immunocyte exosomes also impact vascular biology and disease. CD4+ T cell exosomes enhanced cholesterol accumulation and induced the production of the proinflammatory cytokine TNF-α in cultured human monocytes, which suggests that T cell exosomes might play a role in the pathogenesis of atherosclerosis [120]. T-cell-derived exosomes have been shown to contain thrombospondin-1 and its receptor CD47, which promote endothelial cell responses to vascular endothelial growth factor (VEGF) and accelerate tube formation via VEGF signaling, which suggests a role for exosomes in CD47-dependent tumor angiogenesis [67]. In addition, Aharon, et al. have shown that monocyte-derived exosomes induced procoagulant production and apoptosis of endothelial cells, and accelerated endothelial thrombogenicity, consistent with a role in endothelial cell dysfunction that is associated with inflammatory diseases [121]. 

While exosomes can play an important role in antigen presentation and lymphocyte activation, they also regulate other fundamental immunologic processes. In several studies, cytokines have been detected in immunocyte exosomes, which impact inflammatory responses, innate immunity, and lymphocyte development. Gao, et al. have shown that Dex contain TNF-α, which may play a role in endothelial inflammation and atherosclerosis via membrane TNF-α- induced NF-κB activation [122]. Membrane-bound IL-1β was found on exosomes that were secreted from DCs and macrophages and may play a role in inflammation [123,124]. Wang, et al. reported that TGF-β-containing exosomes derived from thymic cells skewed T cell development in favor of Foxp3+ Tregs, stimulated the differentiation of CD4+CD25 T cells into effector Tregs and induced their proliferation in vitro and in vivo [125]. Another study showed that macrophage exosomes contain miR-223, which can be efficiently transported to monocytes and induce macrophage differentiation [126].

Some critical NK cell activators, including TNF-α, IL-15Rα, and NKG2D ligands, were also found on the surface of human Dex. It has been shown that Dex promote NK cell proliferation and drive NK activation and IFN-γ secretion [76,127]. Further, exosomes that were derived from human macrophages and DCs contained pro-inflammatory leukotrienes (LTs) as well as LT biosynthesis enzymes, which suggests that they can promote inflammation and play a role in the pathogenesis of asthma [59].

An important component of immunity and response to injury is the migration of effector cells to sites of action in vivo. Chemotactic eicosanoids have been identified in macrophage and DC-derived exosomes, which promoted granulocyte migration and recruitment, suggesting that exosomes play a role in leukocyte trafficking [59]. Conversely, Dex which express the chemokine receptor CCR7 have been shown to home to the spleen in response to splenic chemotactic factors CCL19 and CCL21 [128].

Immunocyte exosomes have also been show to regulate stem cell-dependent tissue regeneration. Exosomes from monocytes treated with LPS up-regulated osteogenic gene expression in mesenchymal stem cells (MSC) and induces their osteogenic differentiation [60]. Similarly, intestinal epithelial homoeostasis depends on Wnt/β-catenin signaling crosstalk between the crypt intestinal stem cells (ISC) and the surrounding stromal cells. Saha, et al. found that Wnt was expressed on macrophage-derived exosomes, which could rescue intestinal stem cells and enhance their survival after radiation injury [129].

In a recent neurological study, environmental enrichment (EE) was shown to play a role in neuroprotection in an exosome-dependent manner. Investigators found that serum-derived immunocyte exosomes that were isolated from EE-treated rats significantly increased myelin content, oligodendrocyte precursor (OPC) and neural stem cell levels, and reduced oxidative stress (OS) in vitro. Micro RNA profiling revealed that miR-219 was enriched in rat EE exosomes, which was necessary and sufficient for promoting OPC differentiation into myelinating cells [130]. Likewise, following the treatment of DCs with IFN-γ, Dex carrying specific microRNAs increased baseline myelination, reduced oxidative stress, and improved remyelination in acute autoimmune lysolecithin-induced demyelination, which illustrates the potential of IFN-γ-Dex in the treatment of multiple sclerosis and other demyelinating syndromes [131].

Immunocyte exosomes have also been shown to aid in the repair of cadiovascular injury. Exosomes from DCs co-incubated with hypoxic, necrotic cardiomyocytes were preferentially recruited to the spleen and taken up by CD4+ T cells in mice. Dex increased TNF-α and IFN-γ production in splenic T cells and improved cardiac function in a murine model of myocardial infarction (MI), which suggests a novel strategy for the treatment of MI using Dex-based therapeutics [132].

Taken together, these findings suggest that immunocyte exosomes might not only function in immune modulation and leukocyte migration, but may also regulate stem cell mobilization, tissue remodeling, and wound healing.

In the context of infectious diseases, immunocyte exosomes have been implicated in the response to and sequelae of both bacterial and viral infections. In an animal model of sepsis, Dex containing milk fat globule EGF-factor-8 (MFG-E8), a macrophage surface protein that mediates phagocytosis of apoptotic lymphocytes, decreased plasma TNF-α and IL-6 levels and attenuated inflammation-induced septic shock [133]. In the life cycle of several viruses, exosomes promote viral infection and transmission. Several studies have demonstrated that HIV buds not only from the plasma membrane, but also from lipid rafts in T-cells and macrophages suggesting that viruses can hijack the exosomal pathway to promote viral particle production and dissemination [134,135]. Other studies suggest that human hepatitis C virus might also use a similar strategy to spread in vivo [136].

The consequences of HIV infection also include sequelae that are associated with altered immunocyte exosomes. The treatment of monocytes with IFN-α and LPS to mimic the conditions in the blood of AIDS patients induced the production of exosomes with an altered miRNA profile, which induced the expression of ICAM-1, chemokine ligand (CCL)-2, and IL-6 in endothelial cells [137]. These findings might explain the long-term cardiovascular inflammation in HIV patients. Elucidating the function of exosomes in disease pathogenesis may reveal underlying disease mechanisms and provide targets for the development of therapeutics that block the deleterious effects of exosomes.

## 9. Conclusions and Future Directions

Much effort has been devoted to the development of immunocyte exosomes as therapeutics. They have critical advantages over small molecule pharmacologics and antibody-based agents in that they are directly derived from cell membranes and retain properties of intact cells; they can transfer entire activated signaling pathways and large macromolecular complexes in lipid rafts that present antigens and trigger lymphocyte activation in much the same manner as intact immunocytes. Exosomes can be designed with a high degree of target specificity by engineering them to express ligands for tissue or cell type-specific receptors. Furthermore, their unique lipid composition and resistance to phagocytosis and complement-mediated lysis provide exosomes with protection against degradation in the circulation, which greatly enhances their stability in vivo and protects highly unstable cargo, such as micro RNAs. Exosomes are also stable ex vivo and retain their biological activity after cryogenic storage. Thus, they are uniquely suited for development as therapeutics. Immunocyte exosomes can modulate biological processes as diverse as anti-tumor immunity, stem cell mobilization, and tissue remodeling, which suggests a wide range of clinical applications. Some fundamental challenges to realizing the clinical potential of exosomes still remain, such as maximizing biological function, increasing yield, and controlling their molecular composition. Much of the exosome research over the last several years has focused on overcoming these hurdles.

Dendritic cell-derived exosomes with enhanced biological activity have been generated by direct pulsing of purified exosomes with antigenic peptides, cell lysates, or peptide/MHC complexes, the pre-treatment of DCs with cytokines, such as IFN-γ, to increase the expression of co-stimulatory molecules and pro-inflammatory micro RNAs, ectopic overexpression of antigens, co-stimulatory proteins or adhesion molecules, stimulation of exosome production with TLR4 ligands, and redirected subcellular localization of antigens to the exosome. The enhanced efficacy of Dex can also be achieved by targeting exosomes to secondary lymphoid organs via CCR7-mediated chemotaxis to CCL19 and CCL21 ligands produced by leukocytes in the spleen.

In addition, Dex can also stimulate the innate immune response via NK-activating molecules, such as IL-15 receptor, TNF-α, and NKG2D ligands, which have been shown to activate NK cell proliferation and effector function. Further, loading Dex ex vivo with iNKT ligands, such as galactosylceramide, stimulates pro-inflammatory cytokine production by iNKT cells, which drives proliferation and effector function of NK cells. B and T cell exosomes have been similarly engineered to stimulate immune responses. Covalent binding of C3 complement fragments on B cell exosomes promotes immune activation via interactions with g-protein-coupled receptors on T cells. Likewise, exosomes from IL-2-stimulated T cells potentiate activation of CD3+ T lymphocytes in the absence of APCs. Immunocyte exosomes also package pro-inflammatory and chemotactic mediators that modulate lymphocyte activation and leukocyte migration in vivo. Likewise, exosomes from DCs and macrophages contain the machinery for the biosynthesis of leukotriens and they can be stimulated to produce chemotactic eiconasoids that induce the migration of neutrophils to sites of infection.

In addition to antigen presentation, immunocyte exosomes can also execute effector functions, such as cell-mediate cytotoxicity. Natural killer cell-derived exosomes express NK-related receptors such as NKG2D, FasL, KIR2DL2, as well as cytotoxic factors (perforin, granulysin, granzymes A and B, and TNF-α) that contribute to the lysis and elimination of tumors or virus-infected cells. Likewise, exosomes that are derived from antigen-specific CD8+ CTLs might also target tumor cells for destruction or be used as a delivery system for anti-tumor drugs due to the antigen specificity of their T cell receptors [138].

On the other hand, several studies have shown that immunocyte exosomes can suppress adaptive immunity, suggesting that they might be developed as therapeutics to treat autoimmune diseases or to prevent rejection of organ transplants. Exosomes derived from immature DCs or DCs pre-treated with anti-inflammatory cytokines, such as IL-10 or overexpressing IL-4, can suppress T cell activation and induce immune tolerance. Statin treatment of DCs, which reduces the expression of co-stimulatory molecules, and loading of Dex with TGF-β also have been shown to skew the activity of exosomes toward an immunosuppressive phenotype. Exosomes from DC’s, B, or T cells can induce immunosuppression by various other mechanisms, including Fas-mediated apoptosis and downregulation of MHC-peptide complexes. These studies underscore how knowledge of exosome biogenesis dovetails with the genetic and pharmacologic engineering of more potent exosomes for therapeutic applications.

Immunocyte exosomes have also been shown to regulate processes beyond the immune response. Recent studies have demonstrated the activity of macrophage exosomes on wound healing, osteogenesis, and macrophage development. Likewise, DC-derived exosomes contain osteopontin, chemotactic factors and MMP-9, which may play a role in tissue regeneration mediated by mesenchymal stem cells. Surprisingly, macrophage exosomes can rescue tissue stem cells from radiation injury via Wnt signaling, suggesting that they can be therapeutically used as radioprotectors in cancer therapy or following accidental radiation exposure.

Immunocyte exosomes may also have deleterious pro-inflammatory or tumor-promoting activity. Monocyte exosomes have been shown to possess pro-coagulant properties that might contribute to endothelial cell-mediated thrombosis, T cell exosomes can promote tumor angiogenesis by activating VEGF signaling and induce cancer cell invasion by increasing the production of MMP-9, and exosomes from polarized macrophages can transfer MiRNAs to tumor cells that target anti-metastatic genes. CD4+ T cell exosomes can induce TNF-α in monocytes, which may contribute to atherosclerosis. Likewise, DC exosomes contain TNF-α, which can also contribute to atherosclerotic disease. Thus, immunocyte exosomes represent an unusually versatile class of structures that can be engineered to perform a wide range of functions within and beyond the immune system. However, with this versatility comes the caveat that exosome composition and their interactions with other biological systems must be carefully considered to avoid unwanted, deleterious off-target side effects.

Currently, the development of immunocyte exosomes as vaccines or targeted therapeutics for the treatment of cancer or autoimmunity is still in its early stages. In many trials, exosome vaccines have displayed low activity when compared to intact APCs [93,139]. Possible strategies to improve their efficacy include enhanced exosome purification, loading of more potent antigens, using combinations of antigens, potentiating the vaccines using adjuvants or cytokine treatments, and modulating the efficiency of exosome transport in vivo [43,140]. Exosome-based vaccines can also be used in combination with immunoadujvant therapies that synergistically modulate lymphocyte helper or effector function. It is important to note that immunocyte exosomes in early clinical trials displayed no serious toxicity, which indicates that wider clinical use of exosome-based therapies would likely show minimal side effects.

Recent advances in our understanding of exosome biogenesis raise the intriguing possibility that exosome production and function can be modulated in vivo. The stimulation of P2X7R adenosine receptors on macrophages and DCs increases the production of exosomes expressing the pro-inflammatory mediator IL-1β, synthesis of chemotactic eiconasoids can be induced in Dex and macrophage-derived exosomes by treatment with Ca+ ionophore and arachadonic acid, the production of immunosuppressive Fas+ T cell exosomes is regulated by the activity of diacyglycerol kinase-alpha, and the overall yield of Dex can be increased by TLR4 activation in DCs. Thus, it might be possible to modulate the composition, yield, and biological activity of immunocyte exosomes in vivo using small molecule pharmacologics that regulate exosome biosynthetic pathways, avoiding the technical pitfalls of production, purification, and manipulation of exosomes ex vivo.

Recent exciting advances have placed the field in a position to combine genetic and pharmacologic enhancement of exosome activity with new technologies that increase their purity and yield. A novel method has been reported for preparing exosomes in increased quantities by forcing cells through a screen mesh with apertures that select for vesicles of the appropriate size [141]. Further, a microfluidic cell culture platform has been described that can produce exosomes of defined composition on a large scale by using a 3D-printed microfluidic chip containing chambers for all steps in the process of exosome production and purification [142]. These technologies might obviate much of the technical issues that are involved in exosome production, such as poor yield, low purity, and molecular heterogeneity, which have frustrated efforts to develop exosomes as therapeutic tools.

The rapid pace of advances in exosome biology and clinical application have brought us to the cusp of a new era in therapeutics with the potential to impact a myriad of diseases and biological processes.

## Figures and Tables

**Figure 1 cancers-11-01557-f001:**
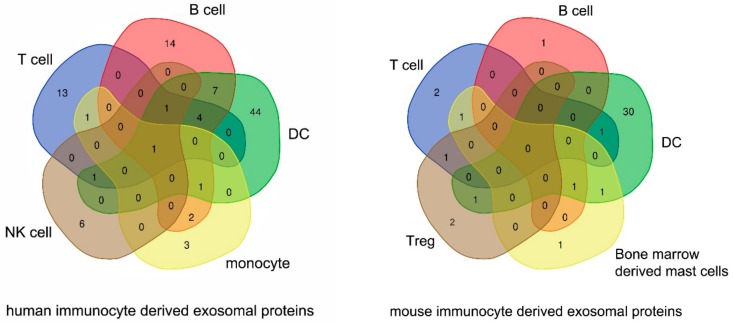
Venn diagram showing the distribution of proteins with immunologic function in immunocyte-derived exosomes from human and murine leukocyte populations. Distribution of proteins common to different immunocyte lineages is shown. Descriptions of the molecules are shown in Table 1.

**Figure 2 cancers-11-01557-f002:**
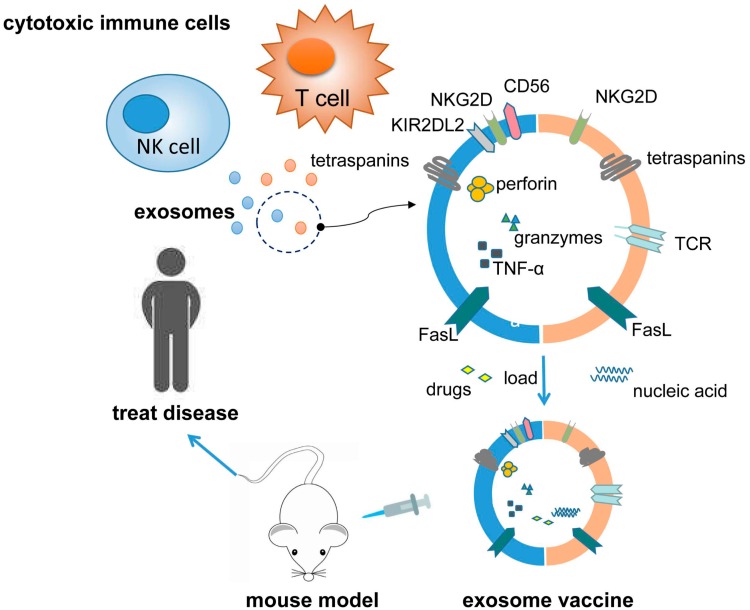
Functions of exosomes derived from cytotoxic immune cells. Exosomes derived from CD8+ T cells and natural killer (NK) cells express recognition receptors for antigens, MHC, or damage signals on target cells and contain cytotoxic mediators, such as perforin, granzymes, and TNF-α that can eliminate tumors or virus-infected targets. These exosomes can also be used as drug delivery systems or vaccines in cancer therapy.

**Figure 3 cancers-11-01557-f003:**
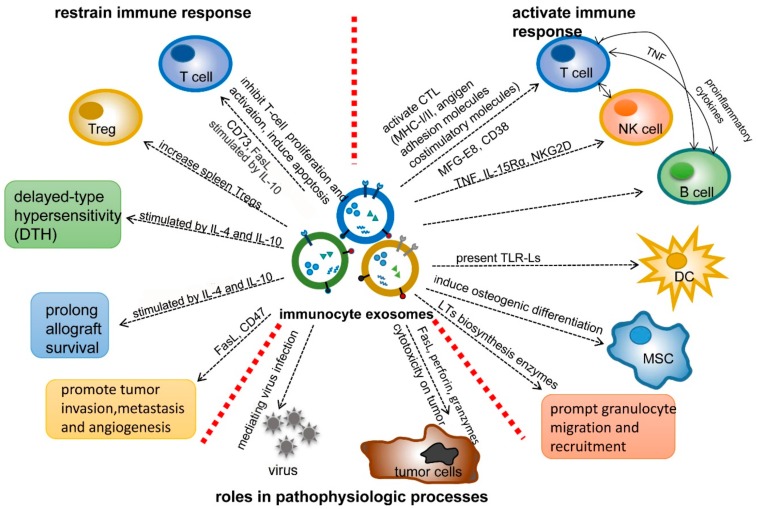
Effects of immunocyte-derived exosomes in the immune response and the pathogenesis of disease. Shown is the spectrum of immunocyte exosome functions that have been identified to date. Immunocyte exosomes stimulate immune responses by potentiating antigen presentation by dendritic cells (DCs) and B cells, driving CD4+ T cell-mediated help, activating cytotoxic effector function of NK and CD8+ T cells, or by directly mediating cytolytic activity. On the other hand, exosomes can be engineered to repress immune function by inhibiting T cell proliferation, inducing lymphocyte apoptosis, and stimulating Treg development and expansion. The composition and function of exosomes can be manipulated by treatment of cells with cytokines, such as IL-4 and IL-10, or expression of molecules such as Fas ligand and CD47. Exosomes are also involved in non-immune functions, such as chemotaxsis and osteogenic differentiation of stem cells and in pathological processes, such as athlerosclerosis, tumor metastasis, autoimmune disease, and sequelae of viral infections.

**Table 1 cancers-11-01557-t001:** Exosomal proteins of human immunocytes.

Exosomal Proteins	Parental Cell Type	Reference
Actin; CD9; CD18 (integrin-β2); CD19; CD20; CD24; CD29(integrin-β1); CD37; CD38; CD45(B220); CD49d(integrin α-4); CD53; CD54(ICAM-1); CD59; CD63; CD71; CD80(B7.1); CD81; CD82; CD86(B7.2); FasL; HLA-DR; Hsp 70; Hsp 90; MHC-I; MHC-II; Moesin; Surface Ig; Tubulin-α; Tubulin-β	B cell	[25,32,37,40,54,55,56]
17 kD fetal brain protein; Actin; ADF; ADP-ribosylation factor 3; AIP-1; Albumine humaine; Alix; Annexin I; Annexin II; Annexin IV; Annexin V; Annexin VI; Arachidonate-15-lipoxygenase; CD1a; CD1b; CD1c; CD1d; CD11c; CD19; CD34; CD40; CD41; CD54; CD59; CD61; CD63; CD80; CD81; CD86; CR3; Cyclophilin A; Cytovillin-2; Enolase; Enzymes for LT biosynthesis; Ezrin; FasL; MFG-E8; BAT3; Gelsolin; HLA-DR; Hsp71; Hsp90; IL-15Rα; Integrin; Major vault protein; MHC-I; MHC-II; MMP-9; Moesin; Osteopontin; Peptidyl prolylcis-transisomerase A; Protein GI; Pyruvate kinase M1; Rab7; Rab-GDP dissociation inhibitor; Ral A; Rap-1b; Cofilin19; Thioredoxin peroxidase 2	Dendritic cell	[57,58,59,60,61,62,63]
Actin; Alix; c-Cbl; CD2; CD3; CD47; CD63; CXCR4; Elongation factor α1; FasL; Glyceraldehyde 3-phosphate dehydrogenase; Hsp90; LFA-1; MHC-I; MHC-II; TCR-β; TCR-ζ; Thrombospondin-1; Tsg101; Tubulin; Tyrosine kinases of the Src family	T cell	[64,65,66,67]
Active tissue factor; CD14; CD18; CD63; CD81; CD9; Hsp70; Tsg101	monocyte	[60,68]
Alix; CD56; CD63; FasL; Fibronectin;Granulysin; Granzymes A; Granzymes B; Perforin	NK cells	[69,70,71]

**Table 2 cancers-11-01557-t002:** Exosomal proteins of mouse immunocytes.

Exosomal Proteins	Parental Cell Type	Reference
C3-fragments; MHC-II	B cell	[25,47]
Alix;Annexins;CCR7;CD9;CD11c;CD40;CD54;CD71; CD80;CD86; Cofilin; DC-SIGN molecules; DEC-205;TLR4;Elongation factor 1α; FasL; Galectin-3; Heteromeric G protein Gi2α; Hsp73; Integrin; Mac-1;MFG-E8;MHC-I;MHC-II;MyD88; PD-L2;Profilin I; Rab 7; Rab 11;Rap 1b; Syntenin; Thioredoxin peroxidase II; TLR9; TNF; TRAIL	Dendritic cell	[27,29,44,48,72,73,74,75,76]
CD54; CD86; LFA-1; MHC-II	Bone marrow-derived mouse mast cell	[77]
CD25; CD4; FasL; LFA-1;TCR	T cell	[78]
CD25; CD9;GITR;LAMP-1	Treg cell	[79]

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
