# Peer review of "Exosomes: Versatile Nano Mediators of Immune Regulation"

_cancers, 2019, doi:10.3390/cancers11101557_

Round 1
Reviewer 1 Report
In this review, authors summarize the function of immune cell derived eoxosme functions, however I would suggest the authors change to EV instead of exosome. Most papers authors cited are EV papers even though they may use exosome as the title.
If authors want to summarize exosome only function, then authors need compare the diffrent type of EVs' functions first.
Author Response
We thank the reviewer for his/her insights and agree that we include papers addressing the function of non-exosomal vesicles, and these need to be treated separately. We have identified five papers in our bibliography that address non-exosomal vesicles.
While we believe that it is of value to discuss these structures as well, this would require, as the reviewer suggests, comparing the similarities and differences between exosomes and other EVs, specifically differences in biogenesis, composition, and function.
However, due to space limitations for this article, we believe that such a discussion is beyond the scope of our manuscript. Thus, we have removed the eight citations that deal with primarily non-exosomal vesicles, and we have also removed all text referring to those papers as well. The remaining papers that we cite focus on vesicles that meet the standard definition of exosomes as described in the literature. A list of the references that have been removed from our bibliography is shown below:
Smyth, L.A.; Ratnasothy, K.; Tsang, J.Y.; Boardman, D.; Warley, A.; Lechler, R.; Lombardi, G. CD73 expression on extracellular vesicles derived from CD4+ CD25+ Foxp3+ T cells contributes to their regulatory function. Eur J Immunol 2013, 43, 2430-2440, doi:10.1002/eji.201242909. Niu, C.; Wang, X.; Zhao, M.; Cai, T.; Liu, P.; Li, J.; Willard, B.; Zu, L.; Zhou, E.; Li, Y., et al. Macrophage Foam Cell-Derived Extracellular Vesicles Promote Vascular Smooth Muscle Cell Migration and Adhesion. J Am Heart Assoc 2016, 5, doi:10.1161/JAHA.116.004099. Aiello, S.; Rocchetta, F.; Longaretti, L.; Faravelli, S.; Todeschini, M.; Cassis, L.; Pezzuto, F.; Tomasoni, S.; Azzollini, N.; Mister, M., et al. Extracellular vesicles derived from T regulatory cells suppress T cell proliferation and prolong allograft survival. Sci Rep 2017, 7, 11518, doi:10.1038/s41598-017-08617-3. Silva, A.M.; Almeida, M.I.; Teixeira, J.H.; Maia, A.F.; Calin, G.A.; Barbosa, M.A.; Santos, S.G. Dendritic Cell-derived Extracellular Vesicles mediate Mesenchymal Stem/Stromal Cell recruitment. Sci Rep 2017, 7, 1667, doi:10.1038/s41598-017-01809-x. Lu, M.; Xing, H.; Xun, Z.; Yang, T.; Zhao, X.; Cai, C.; Wang, D.; Ding, P. Functionalized extracellular vesicles as advanced therapeutic nanodelivery systems. Eur J Pharm Sci 2018, 121, 34-46, doi:10.1016/j.ejps.2018.05.001.
Accordingly, we have removed all text related to these references, as listed below:
page 7, paragraph 2, line 1 “The ecto-5-nucleotidease CD73 is thought to play a role in Treg exosome-mediated immunological tolerance of tumors by converting adenosine-5-monophosphate to adenosine, which drives the production of TGF-b in both target cells and lymphocytes.”
page 11. Paragraph 1. Line 1, “In a recent study, exosomes from athlerosclerosis patients displayed increased ability to promote vascular smooth muscle cell (VSMC) adhesion and migration compared to those of healthy controls. The authors further demonstrated that exosomes from macrophage foam cells promoted aberrant VSMC migration and adhesion by causing the phosphorylation of ERK and Akt in VSMCs, accelerating plaque formation”
Page 11. Paragraph 2, line 12 “Many miRNAs (miR-503, miR-330, miR-293, miR-297c, miR-207, miR-9, miR-484), as well as iNOS were detected in exosomes derived from CD4+CD25- Tregs, which can block cell cycle progression of T cells, induce their apoptosis, and convert naive T cells to Tregs”.
page 11, paragraph 4, line 6 “Likewise, Silva, et al. reported that Dex were also enriched in two chemotactic mediators, osteopontin and MMP-9 which contribute to DC exosome-mediated mesenchymal stem/stromal cells (MSC) recruitment in vitro.”
page 13, paragraph 3, line 9 “Likewise, Silva, et al. reported that Dex were also enriched in two chemotactic mediators, osteopontin and MMP-9 which contribute to DC exosome-mediated mesenchymal stem/stromal cells (MSC) recruitment in vitro.”
We would like to emphasize that our revisions do not in any way change the basic arguments in our manuscript, the flow of the text, or our conclusions.
We hope that our revisions satisfactorily address this concern.
Reviewer 2 Report
This is a nice comprehensive review with several examples relevant to the immunomodulatory capacity of EVs.
A minor comment page 1 is the "inward budding of the Golgi network to generate exosomes" correct?
Author Response
Reviewer 2.
We appreciate the reviewer’s positive comment that our manuscript is a “nice comprehensive review”
This reviewer had one minor comment, which we address below:
Comment:
“A minor comment on page 1 is the ‘inward budding of the Golgi network to generate exosomes’ correct?”
Response:
We thank the reviewer for his/her comment. We have revised this statement in the text. Indeed, exosomes do not form from budding of the Golgi. However, protein cargo is often loaded into exosomes via a trans-Golgi network. We have removed the original statement and replaced it with the more correct description of protein loading via the trans-Golgi mechanism. This statement is highlighted in bold in the revised manuscript.
Reviewer 3 Report
Comments:
Explain the mechanism for exosome capture and internalization. It is very important in the context of this review
Figure 1 is very confusing, and probably false as it is shown. I do not believe that, for example, exosomes from NK and B only have 2 proteins in common. They should have in common at least the characteristics exosome proteins.
In figure 3 it seems that exosomes carry cytokines, but it is not explained in the text.
In general bibliography is quite old. There are few citations for manuscripts from 2018-19
Author Response
Reviewer 3.
We thank the reviewer for his/her comments. Please see our responses below.
Comment 1:
“Explain the mechanism for exosome capture and internalization”
Response:
We have added a more detailed discussion regarding the current understanding of exosome binding, entry, and release of cargo into target cells and comment on the importance of understanding this process for the design of exosomes as targeted therapies and for the processing of exosomal antigens by APCs for presentation to lymphocytes and activating the immune response. The revisions appear in the second and third paragraphs of the introduction on page 2 of the manuscript, and is highlighted in bold.
We hope that our revisions adequately addresses this concern.
Comment 2:
“Figure 1 is very confusing and probably false as it is shown. I do not believe that, for example, exosomes from NK and B only have 2 proteins in common”
Response:
We apologize for any confusion regarding Figure 1. This figure (and Tables I and II) are primarily designed to emphasize only those proteins common to different immunocyte lineages that have immunologic function, as these are the most relevant to our discussion. We feel that adding all other common proteins, such as those involved in basic exosome biogenesis would be confusing as they are the most abundant and most widely shared among exosomes from different cellular origins, and would make it difficult to focus on those proteins that are the specific focus of our manuscript. We have added clarification in the legend for Figure 1 to avoid confusion on the part of the reader.
Comment 3:
“In figure 3 it seems that exosomes carry cytokines, but it is not explained in the text”
Response:
We apologize for any confusion. The reference to cytokines (IL-4 and IL-10) in figure 3 does not signify molecules that are carried by exosomes, but rather indicates cytokine treatments of cells that modulate exosome composition and function. We have added clarification in Figure 3 as well as the figure legend.
Comment 4:
“In general bibliography is quite old.”
Response:
We thank the reviewer for his/her observations. In order to make our review more current, we have added several recent citations and added supporting text, which is highlighted in bold in the revised manuscript. The following citations have been added to our manuscript:
Hessvik NP, Llorente A (2018) Current knowledge on exosome biogenesis and release. Cell Mol Life Sci. Jan;75(2):193-208.
Kwon SH, Oh S, Nacke M, Mostov KE, Lipschutz JH (2017) Adaptor protein CD2AP and L-type lectin LMAN2 regulate exosome cargo protein trafficking through the Golgi complex J Biol Chem. Oct 6;292(40):16523
Mathieu M, Martin-Jaular L, Lavieu G, Théry C (2019) Specificities of secretion and uptake of exosomes and other extracellular vesicles for cell-to-cell communication. Nat Cell Biol. Jan;21(1):9-17.
Kelly J. McKelvey, Katie L. Powell, Anthony W. Ashton, Jonathan M. Morris, and Sharon A. McCracken (2015) Exosomes: Mechanisms of Uptake J Circ Biomark. Jan-Dec; 4: 7.
102.Hiltbrunner S, Larssen P, Eldh M, Martinez-Bravo MJ, Wagner AK, Karlsson MC, Gabrielsson S (2016) Exosomal cancer immunotherapy is independent of MHC molecules on exosomes. Oncotarget. Jun 21;7(25):38707-38717
Samuel M, Gabrielsson S (2019) Personalized medicine and back-allogeneic exosomes for cancer immunotherapy. J Intern Med. Jul 29. doi: 10.1111/joim.12963. [Epub ahead of print]
108.Wang X, Shen H, He Q, Tian W, Xia A, Lu XJ (2019) Exosomes derived from exhausted CD8+ T cells impaired the anticancer function of normal CD8+ T cells J Med Genet. Jan;56(1):29-31.
Shu-Wei Wu, Lei Li, Yan Wang, and Zhengguo Xiao (2019) CTL-Derived Exosomes Enhance the Activation of CTLs Stimulated by Low-Affinity Peptides Front Immunol. 10: 1274.
Gehrmann U, Näslund TI, Hiltbrunner S, Larssen P, Gabrielsson S (2014) Harnessing the exosome-induced immune response for cancer immunotherapy Semin Cancer Biol. Oct;28:58-67.
Round 2
Reviewer 1 Report
Authors already corrected all the comments.
Reviewer 3 Report
The authors have correctly addressed all my concerns, and consequently I recommend this manuscript for publication in this journal